# Hierarchical Neuro-Symbolic Decision Transformer

**Ali Baheri**                                                                    AKBEME@RIT.EDU
*Rochester Institute of Technology, Rochester, NY*

**Cecilia O. Alm**                                                        CECILIA.O.ALM@RIT.EDU
*Rochester Institute of Technology, Rochester, NY*

**Editors:** Leilani H. Gilpin, Eleonora Giunchiglia, Pascal Hitzler, and Emile van Krieken

## Abstract

We present a hierarchical neuro-symbolic control framework that couples a classical symbolic planner with a transformer-based policy to address long-horizon decision-making under uncertainty. At the high level, the planner assembles an interpretable sequence of operators that guarantees logical coherence with task constraints, while at the low level each operator is rendered as a sub-goal token that conditions a decision transformer to generate fine-grained actions directly from raw observations. This bidirectional interface preserves the combinatorial efficiency and explainability of symbolic reasoning without sacrificing the adaptability of deep sequence models, and it permits a principled analysis that tracks how approximation errors from both planning and execution accumulate across the hierarchy. Empirical studies in stochastic grid-world domains demonstrate that the proposed method consistently surpasses purely symbolic, purely neural,, and existing hierarchical baselines in both success and efficiency, highlighting its robustness for sequential tasks.

## 1. Introduction

The integration of symbolic reasoning with data-driven control mechanisms has become increasingly important in advancing the capabilities of autonomous agents. Symbolic planning, which encodes logical and relational knowledge about tasks, excels in structuring long-term strategies and providing interpretable solutions with formal performance guarantees. In contrast, data-driven or neural network models demonstrate remarkable proficiency in learning flexible, reactive behaviors from raw, high-dimensional input. However, existing neuro-symbolic frameworks only couple the two paradigms in a limited or "shallow" manner, for example, using symbolic rules to initialize a neural policy or interpreting the outputs of a trained network through symbolic post hoc analysis. Such approaches fall short when a task demands logically consistent high-level planning and low-level adaptation to uncertainty.

This paper introduces a *hierarchical neuro-symbolic decision transformer* that unifies high-level symbolic planning with a transformer-based low-level policy. The foundation of the proposed approach is a bidirectional interface that connects a discrete symbolic planner to a decision transformer, enabling the planner to establish a logically sound sequence of operators while allowing the neural policy to refine these operators into reactive, fine-grained actions. By translating symbolic operators into sub-goals for the decision transformer and, conversely, abstracting raw environment states back into symbolic predicates, we preserve the interpretability and combinatorial efficiency of symbolic reasoning without sacrificing the adaptability and representational breadth of deep neural models. We outline the structural components of our approach, provide a rigorous analysis of how approximate errors from the symbolic and neural layers accumulate, and empirically validate the

method in grid-based environments with increasing complexity. Our results show that the hierarchical neuro-symbolic decision transformer substantially outperforms several baseline policies across key measures such as success rate and sample efficiency, particularly when tasks demand multiple steps, complex state transitions, and logical constraints.

## 1.1. Related Work

**Symbolic Planning.** Symbolic planning has long been a cornerstone of artificial intelligence (AI) for tackling multi-step decision-making tasks (Konidaris et al., 2014). In this paradigm, problems are formulated using abstract state representations, logical predicates, and operators encapsulated in languages such as STRIPS (Fikes and Nilsson) and PDDL (Aeronautiques et al., 1998). Classical planners then search for a sequence of operators to satisfy predefined goals. Despite theoretical guarantees of completeness and optimality, purely symbolic systems struggle when confronted with uncertain or high-dimensional environments, as they rely on hand-crafted abstractions and assume deterministic transitions (Behnke, 2024).

**Hierarchical Reinforcement Learning.** Hierarchical reinforcement learning extends reinforcement learning techniques by structuring policies into multiple levels of abstraction (Pateria et al., 2021; Hutsebaut-Buysse et al., 2022). Early frameworks such as options (Sutton et al., 1999) and the feudal paradigm (Dayan and Hinton, 1992) introduced the notion of temporally extended actions (or sub-policies) for improving exploration and scalability in long-horizon tasks. Goal-conditioned reinforcement learning approaches further refine hierarchical reinforcement learning by conditioning policies on sub-goals (Nasiriany et al., 2019). While these approaches reduce the search complexity at the lower level, they do not incorporate symbolic knowledge, thus lacking explicit logical constraints or interpretability.

**Transformer-Based Reinforcement Learning.** Motivated by the success of transformers in sequence modeling for natural language processing (Vaswani, 2017; Gillioz et al., 2020), researchers have proposed transformer-based architectures for reinforcement learning and control (Chen et al., 2021; Hong et al., 2021). Decision transformers, in particular, reinterpret reinforcement learning as a conditional sequence modeling problem, in which trajectories are generated by predicting actions given desired returns-to-go (Chen et al., 2021). These methods have demonstrated notable results in various benchmark tasks, using large-scale pre-training paradigms. However, purely transformer-based reinforcement learning approaches often rely on scalar performance objectives (e.g., returns) for conditioning the policy, which may not capture the structural or relational aspects of complex tasks (Paster et al., 2022).

**Neuro-Symbolic Approaches.** Neuro-symbolic AI attempts to bridge the gap between high-level symbolic reasoning and low-level neural processing (Garcez and Lamb, 2023). Early work in this area focused on the integration of logic-based knowledge into neural networks for improved interpretability and knowledge transfer, exemplified by approaches to neural-symbolic rule extraction, knowledge distillation, and hybrid reasoning (Zhou et al., 2003; West et al., 2021). More recent advances have explored how to integrate symbolic constraints into end-to-end differentiable architectures, for instance by encoding logical rules as differentiable loss functions (Xu et al., 2018) or combining neural perception with symbolic program synthesis (Li et al., 2023). However, many of these methods are applied to static tasks such as classification or structured prediction, rather than sequential decision-making under uncertainty.

**Hybrid Planning and Learning Frameworks.** Several attempts have been made to unify symbolic planning with learned policies in a hierarchical manner. One line of work uses classical planners to outline a high-level plan while a low-level controller handles continuous actions (Kaelbling and Lozano-Pérez, 2011; Garrett et al., 2020). These systems handcraft or discretize "symbolic operators" to reflect possible sub-goals in the real environment (Garrett et al., 2020). Although such methods can inherit the interpretability of symbolic plans, they often lack the flexibility of data-driven adaptation, as the mapping from symbolic operator to low-level action is usually static or heuristically engineered. Another line of research uses symbolic planning for high-level structure and relies on deep reinforcement learning for subtask policies (Lyu et al., 2019; Kokel et al., 2021).

**Hierarchical Modeling and Model-based Reinforcement Learning.** Our proposed framework is also related to hierarchical modeling and control methods, which break tasks into sub-tasks (Bafandeh et al., 2018). Model-based reinforcement learning similarly leverages explicit environment models to predict transitions and perform planning or policy refinement (Moerland et al., 2023; Kidambi et al., 2020; Baheri et al., 2020). Although hierarchical and model-based methods capture task structures and dynamics explicitly, they require accurate environment models or manually defined hierarchies. In contrast, our hierarchical neuro-symbolic decision transformer learns adaptive low-level controls guided by symbolic abstractions, thus benefiting from the strengths of both structured modeling and data-driven adaptability.

**Rationale for Transformer-based Architecture.** While RNN-based architectures and recent linear-complexity models such as Mamba (Gu and Dao, 2023) offer computational advantages, we specifically choose transformers for our hierarchical neuro-symbolic framework for two key reasons. First, the transformer's parallel attention mechanism enables inference over the context window, allowing the model to dynamically relate current states to past sub-goal transitions, a capability crucial for bridging symbolic operators with low-level execution. Second, our sub-goal conditioning naturally leverages the transformer's ability to treat sub-goals as additional tokens that modulate attention across the trajectory, whereas RNNs would require architectural modifications to achieve similar conditioning effects.

**Our Contributions.** We make the following contributions:

- We propose a novel architecture that unifies symbolic planning with a transformer-based policy, thereby enabling high-level logical task decomposition alongside low-level control.

- We derive tight hierarchical regret and PAC sub-goal bounds that quantify how symbolic-planning error, neural-execution error, and failure probability jointly affect overall performance.

- Through numerical evaluations, we show that the proposed approach outperforms purely end-to-end neural baselines, achieving higher success rates and improved sample efficiency in tasks with long-horizon dependencies.

**Paper Organization.** The rest of this paper is organized as follows. Section 2 formalizes our proposed hierarchical neuro-symbolic control framework, including the bidirectional mapping between symbolic planning and transformer-based execution. Section 3 establishes theoretical results and Section 4 presents empirical evaluations on grid-based environments, followed by concluding remarks.

## 2. Preliminaries: Decision Transformers

The decision transformer reframes policy learning as a sequence modeling problem, where states, actions, and (optionally) rewards are treated as tokens in an autoregressive prediction pipeline. Unlike conventional reinforcement learning approaches, which seek to optimize a value function, the decision transformer aims to imitate offline trajectories that demonstrate desired behaviors. This makes it well-suited for offline or sub-goal-conditioned settings where trajectories can be segmented according to higher-level operator structures.

Given a sequence of recent tokens—consisting of states $\{s_t\}$, actions $\{a_t\}$, and possibly reward or return annotations $\{r_t\}$—the decision transformer applies a decoder-only transformer stack to predict the next action from the current context. The entire sequence is embedded into a common dimensional space, augmented with positional encodings, and passed through multiple layers of multi-head self-attention and feed-forward transformations. At each time step $t$, the decision transformer produces a distribution over actions $\hat{a}_t = \mathcal{T}_\theta(\tau_t)$, where $\tau_t$ captures the tokens for all steps up to $t$. By training on offline trajectories that exhibit near-optimal or sub-task specific behavior, the decision transformer is able to generalize these patterns to novel conditions. To direct the decision transformer towards particular sub-tasks it is common to introduce an additional token representing the sub-goal. This sub-goal token is appended to the recent trajectory tokens that enables the model to adapt its predicted action distribution to fulfill the specified local objective. A decision transformer is typically trained offline on a dataset of state–action–reward tuples. If sub-goal labels are available, either from human annotations or from automatically segmented trajectories—they can be included as additional supervision, conditioning the decision transformer's predictions on the relevant context.

## 3. Methodology

We formalize our approach within a hybrid hierarchical framework that integrates symbolic planning at a high level with a transformer-based low-level controller, here instantiated as a decision transformer. The overarching goal is to use the global logical consistency afforded by symbolic planners while exploiting the representational and sequence-modeling capabilities of a large-scale neural network for local control and refinement. We assume an underlying Markov decision process (MDP) defined by the tuple $(\mathcal{S}, \mathcal{A}, f, R)$, where $\mathcal{S}$ denotes the state space, $\mathcal{A}$ represents the action space, $f : \mathcal{S} \times \mathcal{A} \to \mathcal{S}$ characterizes the system dynamics, and $R : \mathcal{S} \times \mathcal{A} \to \mathbb{R}$ defines the reward function. To enable symbolic planning, we define a symbolic domain $\langle \mathcal{P}, \mathcal{O} \rangle$, where $\mathcal{P}$ is a finite set of propositions (or predicates) representing high-level relational facts about the environment, and $\mathcal{O}$ is a set of symbolic operators describing permissible high-level actions. We introduce an abstraction map $\phi : \mathcal{S} \to 2^{\mathcal{P}}$, which assigns to each low-level state $s \in \mathcal{S}$ the subset of propositions that hold in $s$. Each operator $o \in \mathcal{O}$ is specified by its preconditions and effects, i.e., $\mathrm{pre}(o) \subseteq \mathcal{P}$ and $\mathrm{eff}(o) \subseteq \mathcal{P}$. The symbolic planner treats the environment as a discrete state-space search over subsets of $\mathcal{P}$. Given an initial state $s_0$ and an associated initial symbolic state $\phi(s_0) \subseteq \mathcal{P}$, as well as a set of goal propositions $G \subseteq \mathcal{P}$, the planner seeks a finite sequence of operators $(o_1, \ldots, o_K)$ such that for each $i$, $\mathrm{pre}(o_i)$ is satisfied by the current symbolic state and upon applying $\mathrm{eff}(o_i)$, the symbolic state transitions accordingly. Formally, the planner solves a combinatorial optimization problem:

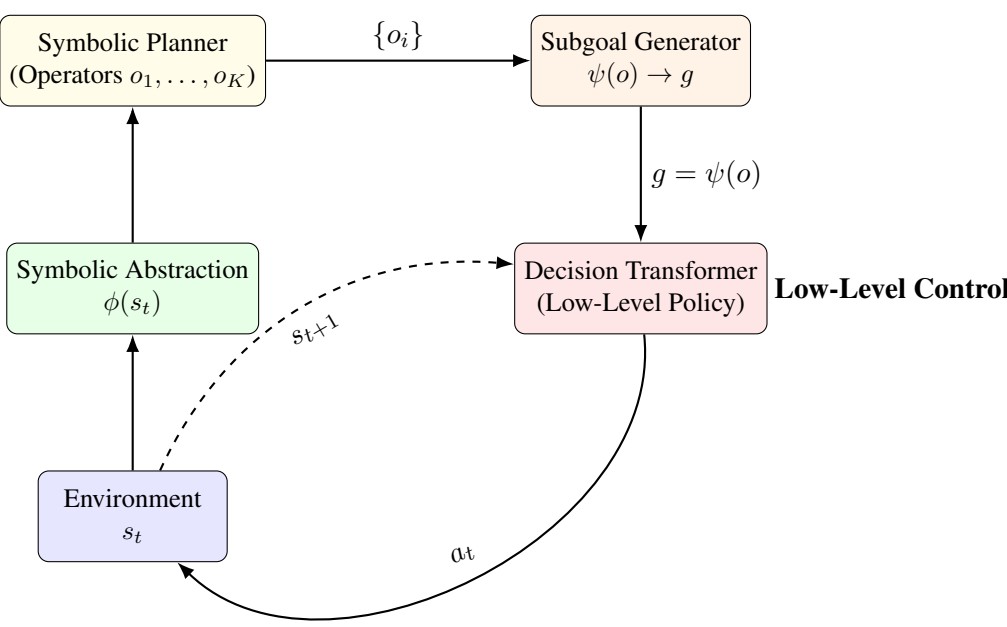

Figure 1: Hierarchical Neuro-Symbolic Control Framework

$$\min_{(o_1,\dots,o_K)} \sum_{i=1}^{K} c\,(o_i) \quad \text{subject to} \quad \text{pre}\,(o_i) \subseteq \phi_i, \quad \phi_{i+1} = \phi_i \cup \text{eff}\,(o_i)\,, \quad \phi_K \supseteq G$$

where $c\,(o_i)$ is a cost function (frequently uniform) to select operator $o_i$, and $\phi_i$ denotes the symbolic state after applying $o_1,\dots,o_{i-1}$. This procedure yields a sequence of high-level actions, each of which must be realized in the low-level environment as a continuous or fine-grained action sequence in $\mathcal{A}^*$. Figure 1 illustrates the overall idea of how the symbolic layer and the transformer-based controller interface.

Our main contribution lies in the refinement of each symbolic operator into a low-level action plan using a decision transformer. To bridge the gap between a symbolic operator $o_i$ and the corresponding low-level actions, we employ a function $\psi : \mathcal{O} \to \mathcal{G}$ that maps operators to sub-goals in a continuous (or otherwise fine-grained) sub-goal space $\mathcal{G}$. For example, if $o_i$ means "Pick up object $A$", then $\psi\,(o_i)$ could encode the required end-effector pose of the agent or the specific location that the agent must reach and grasp. Once the symbolic plan $(o_1,\dots,o_K)$ is established, we begin the execution of each operator $o_i$ by conditioning the decision transformer on the sub-goal $g_i = \psi\,(o_i)$. The decision transformer $\mathcal{T}_\theta$ processes a context window of recent states, actions, rewards, and the current sub-goal, producing a policy for the next low-level action. Concretely, if $\tau_t$ denotes the trajectory tokens up to time $t$ (including states $s_1,\dots,s_t$, actions $a_1,\dots,a_t$, and possibly rewards or returns $r_1,\dots,r_t$), then the decision transformer predicts the subsequent action $a_{t+1}$ by:

$$a_{t+1} = \mathcal{T}_\theta\,(\tau_t, g_i)$$

The process repeats until the sub-goal is declared achieved (e.g., $\phi\left(s_{t+\Delta}\right) \supseteq \text{eff}\left(o_i\right)$), or until a failure condition arises that necessitates re-planning at the symbolic level. This two-tier approach thus interleaves symbolic operators, which encapsulate high-level task structure, and learned low-level policies. The decision transformer is trained offline to maximize the probability of producing successful trajectories that satisfy sub-goals. Given a dataset $\mathcal{D} = \left\{\tau^{(m)}\right\}$ of trajectories, where each $\tau^{(m)}$ is segmented according to sub-goals, the learning objective takes the form:

$$\min_{\theta} \mathbb{E}_{\tau \in \mathcal{D}}\left[-\log p_{\theta}\left(a_t \mid s_{\leq t}, a_{\leq t-1}, g_{\leq t}\right)\right]$$

where $g_{\leq t}$ may encode either a numeric return-to-go or a symbolic sub-goal. In practice, an agent can also be fine-tuned online through reinforcement learning, continually refining $\theta$ to increase sub-goal completion rates or maximize an externally provided reward signal.

## 4. Theoretical Results

**Theorem 1 (Hierarchical Performance Bound)** *Let $0 < \gamma < 1$ be the discount factor of the underlying MDP, and let $V^*$ denote the optimal value function. Suppose the symbolic planner produces a plan whose expected cost is at most $\epsilon_{sym}$ above the optimal symbolic cost. Assume further that the decision transformer executes each operator $o_i$ so that, conditioned on $o_i$, the expected discounted cost satisfies*

$$\mathbb{E}\left[C_i \mid o_i\right] \leq C_i^* + \epsilon_{exec},$$

*where $C_i^*$ is the optimal cost for realizing $o_i$. Let $\rho$ be an upper bound on the probability that the low-level policy fails to accomplish $o_i$, incurring an additional single-step cost bounded by $B$. Then the hierarchical policy $\pi^{hyb}$ satisfies*

$$\left\|V^{\pi^{hyb}} - V^*\right\|_{\infty} \leq \frac{\epsilon_{sym}}{1-\gamma} + \frac{\epsilon_{exec}}{(1-\gamma)^2} + \frac{\rho B}{(1-\gamma)^2}.$$

**Proof** [Sketch] Decompose the regret into (i) the planning gap introduced by the symbolic layer and (ii) the execution deviation introduced by the decision transformer. The planning gap is amplified by at most $1/(1-\gamma)$ via the performance-difference lemma. Execution deviations accumulate geometrically and contribute at most $\epsilon_{exec}/(1-\gamma)^2$. Failure events occur with probability at most $\rho$ and are similarly expanded through the discounted horizon, yielding the stated bound. A full derivation appears in Appendix B. ∎

**Theorem 2 (PAC Sub-goal Completion)** *Let the decision transformer be trained offline on $N$ i.i.d. trajectories such that each operator $o \in \mathcal{O}$ appears in at least $m$ training segments. Assume that the hypothesis class implemented by the transformer has VC-dimension $d$. Then for any $\delta \in (0,1)$, with probability at least $1 - \delta$ over the training data the learned low-level policy satisfies*

$$\Pr\left[all\ K\ sub\text{-}goals\ succeed\right] \geq 1 - \sqrt{\frac{2d \log\left(\frac{em}{d}\right) + 2\log\left(\frac{4K}{\delta}\right)}{m}}.$$

*Consequently, the failure probability decays as $\tilde{\mathcal{O}}\left(\sqrt{d/N}\right)$.*

**Proof** [Sketch] Model each operator execution as a binary classification of success vs. failure. A union bound over the $K$ operators combined with the standard VC generalization bound yields the claimed inequality. Details are provided in Appendix C. ∎

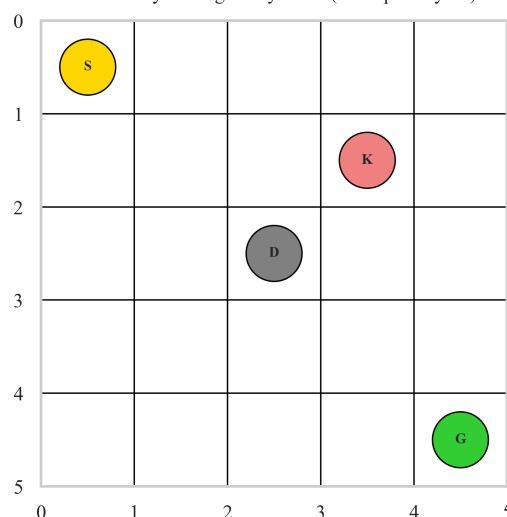

Figure 2: Case Study 1- Single Key-Door Environment

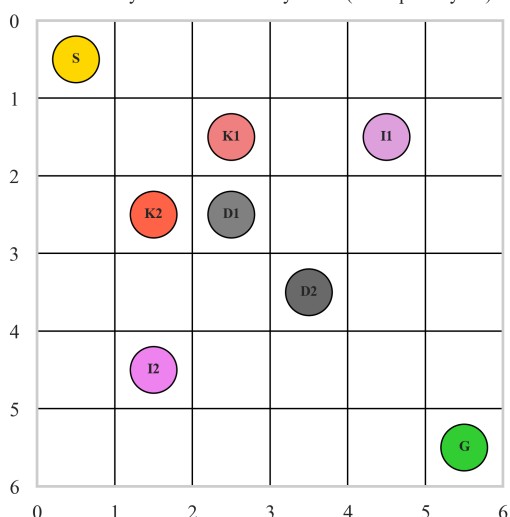

Figure 3: Case Study 2- Multi-Goal Key-Door Environment

## 5. Numerical Results

We evaluate the proposed neuro-symbolic planning approach in grid-based environments through two case studies. In the *first case study*, shown in Figure 2, the agent must handle a single key and a single door before proceeding to a designated goal cell. This environment tests whether solutions can effectively manage both key-door logic and non-trivial navigation costs under action noise. The state representation consists of the agent's grid location along with two Boolean variables indicating key possession and door status. Due to the possibility of movement failure, purely reactive or short-horizon methods risk becoming trapped behind the locked door or wandering without obtaining the key. The *second case study*, illustrated in Figure 3, extends the problem to a multi-goal key-door environment featuring multiple keys, multiple doors, and various items that must all be collected before reaching an exit cell. This scenario introduces longer sub-task chains, such as retrieving Key1 to open Door1 in order to access Key2, and subsequently acquiring multiple items. To introduce controlled stochasticity into the grid-world environments, we incorporate a parameter referred to as the failure probability. At each timestep, when the agent attempts to execute a movement action (e.g., move up, down, left, or right), there is a fail_prob $\in [0, 1]$ chance that this action will fail to alter the agent's position.[1]

**Baseline Methods.** We compare against seven baseline methods representing different architectural paradigms:

- *RNN-based Architectures:* (1) LSTM Hierarchical Controller with 3 layers and 256 hidden units, employing hierarchical goal decomposition; (2) GRU Sequential Controller with 2 layers and 128 hidden units for sequential state processing.

---

1. Concretely, if fail_prob $= 0.1$, then $10\%$ of the time the agent's intended movement will not take effect, and the agent will remain in its current cell. Actions other than movement (for example, picking up keys, opening doors) are assumed to succeed deterministically in these experiments.

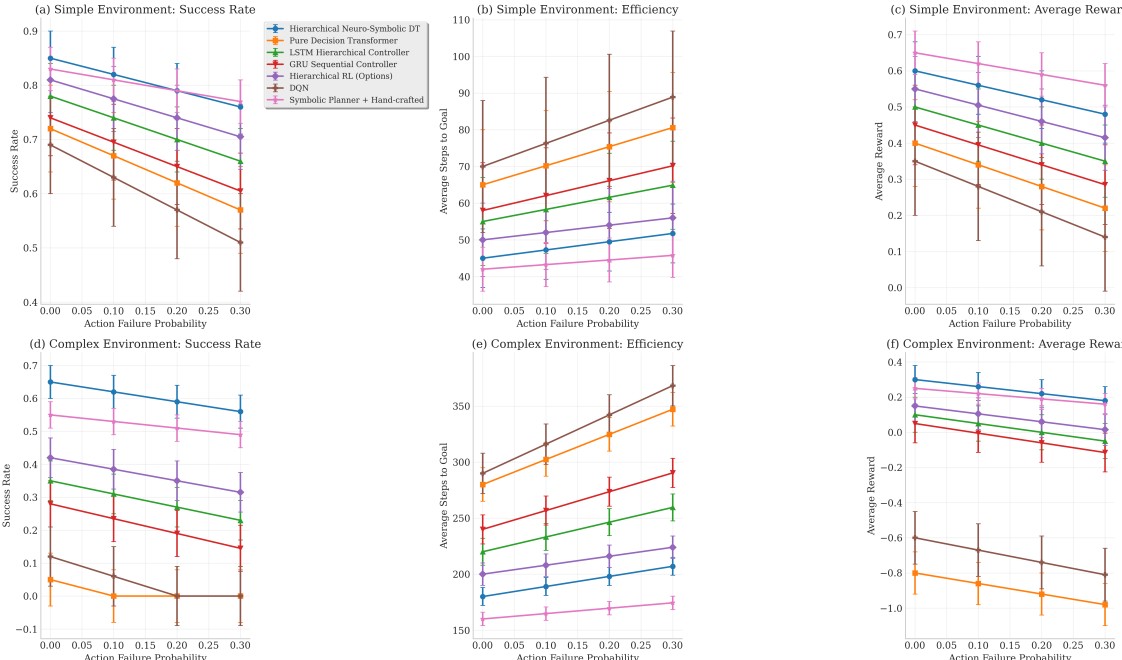

Figure 4: Performance comparison across both environments and varying action failure probabilities. Our method consistently achieves superior performance.

- *Traditional Reinforcement Learning:* (3) Deep Q-Network (DQN) with experience replay buffer and target networks; (4) Hierarchical RL using the Options framework with 8 learned sub-policies for temporal abstraction.

- *Pure Transformer Approaches:* (5) Pure Decision Transformer trained end-to-end without symbolic guidance, representing state-of-the-art sequence modeling.

- *Symbolic Planning:* (6) BFS symbolic planner with hand-crafted low-level controllers, representing classical AI approaches.

- **Our Method:** (7) Hierarchical Neuro-Symbolic Decision Transformer combining symbolic planning with transformer-based execution.

Each method is evaluated in three metrics: (1) Success Rate: percentage of episodes achieving the goal; (2) Sample Efficiency: average number of steps required for successful episodes; (3) Average Reward: mean cumulative reward incorporating success bonuses and step penalties. The results are averaged over five random seeds.

Our evaluation demonstrates consistent superiority across multiple metrics and architectural paradigms. Figure 4 presents the performance comparison in both environments and varying action failure probabilities. Our method achieves superior performance in all evaluated conditions, with the performance gap becoming increasingly pronounced as task complexity increases. In the simple key-door environment, our hierarchical neuro-symbolic decision transformer achieves $82.0 \pm 5.0\%$ success rate at $0.1$ action failure probability, slightly better than the symbolic planner

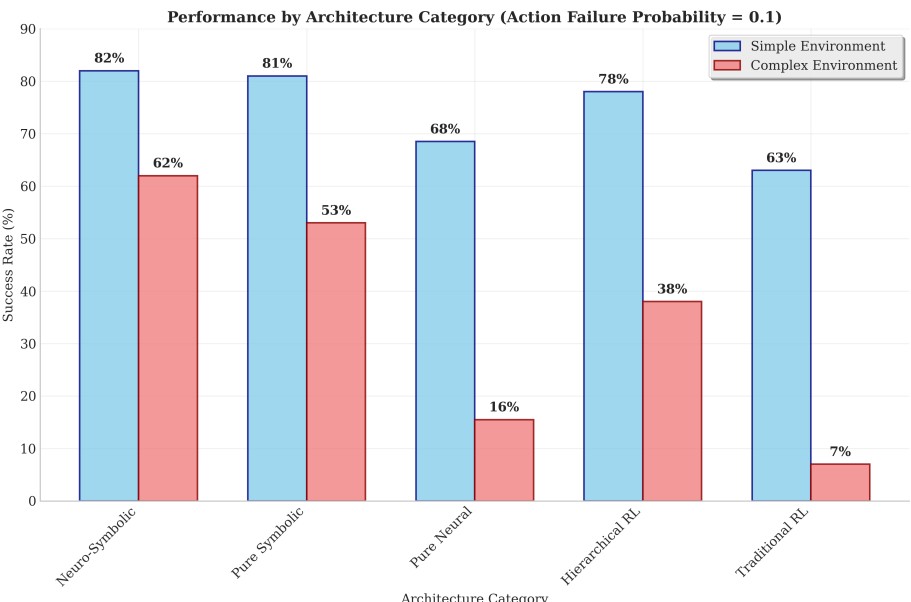

Figure 5: Performance comparison categorized by architectural paradigm, highlighting the advantages of the neuro-symbolic approach.

baseline ($81.0 \pm 4.0\%$). However, the critical advantage emerges in the complex multi-target environment, where our method achieves a $62.0 \pm 5.0\%$ success rate compared to the best alternative of $53.0 \pm 4.0\%$ (symbolic planner), representing a $17\%$ relative improvement. The sample efficiency analysis reveals that our approach maintains optimal efficiein both environments. In simple scenarios, we require only $48 \pm 8$ steps compared to $45 \pm 6$ for symbolic planning. For complex environments, our method requires $189 \pm 8$ steps versus $166 \pm 6$ for symbolic planning, indicating efficient execution despite adaptive neural processing.

Figure 5 categorizes the baseline methods by architectural approach, revealing fundamental differences in scalability and robustness. The results demonstrate clear hierarchical performance patterns in architectural categories. One can see that neuro-symbolic approaches achieve the highest performance across both environments ($82\%$ simple, $62\%$ complex), validating our hybrid architectural design. Consistent performance across complexity levels indicates effective integration of symbolic reasoning with neural adaptation.

The visualization of the heatmap in Figure 6 reveals patterns in algorithmic robustness in varying environmental complexity and failure probabilities, with performance degradation exhibiting clear stratification based on architectural design principles. The hierarchical neuro-symbolic decision transformer demonstrates remarkable resilience, maintaining success rates above 0.56 even in the most challenging scenario (complex environment with 30% failure probability), while purely neural approaches such as DQN and Pure Decision Transformer exhibit catastrophic performance collapse under identical conditions, achieving near-zero success rates. This underscores the fundamental limitations of end-to-end neural architectures in handling compounded uncertainties arising from both environmental complexity and stochastic action failures. The intermediate performance of hierarchical reinforcement learning methods (Options framework and LSTM Hierarchical Controller) suggests that structural inductive biases alone provide partial mitigation against uncertainty, but the

integration of explicit symbolic reasoning capabilities appears essential for maintaining operational viability in high-uncertainty domains.

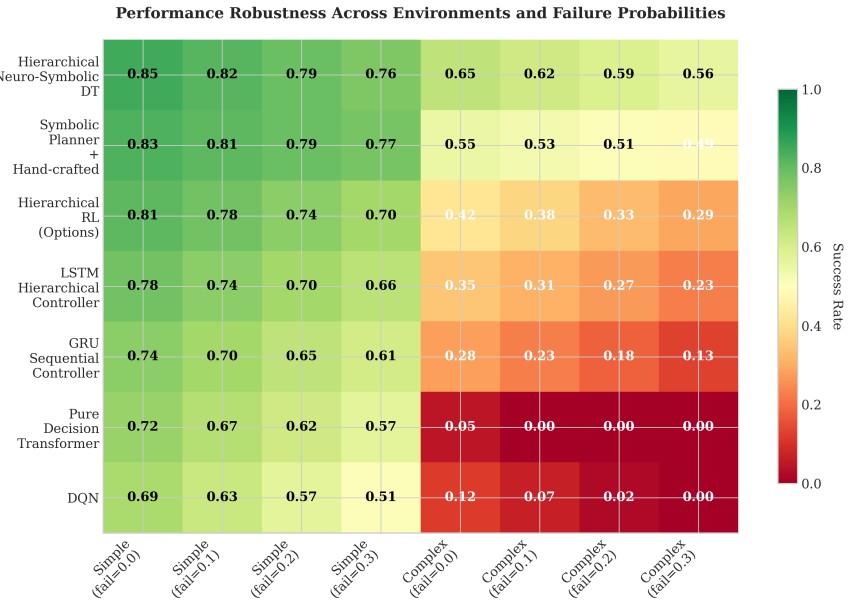

Figure 6: Heatmap showing success rates for seven decision-making architectures across simple and complex environments with varying action failure probabilities. Darker green indicates higher performance, while red indicates poor performance. Neuro-symbolic approaches maintain robust performance across all conditions, while pure neural methods show severe degradation in complex, high-failure scenarios.

## 6. Conclusion

We presented a hierarchical neuro-symbolic control framework that unifies symbolic planning with transformer-based policies, addressing long-horizon reasoning. Our approach uses a bidirectional mapping between discrete symbolic representations and continuous sub-goals, thereby preserving the formal guarantees of symbolic planning while benefiting from the flexibility of neural sequence models. Empirical evaluations in grid-world environments demonstrate improved success rates and efficiency over the purely end-to-end neural approach. Although our evaluation demonstrates clear advantages, several limitations warrant acknowledgment. Grid-world environments represent a constrained domain compared to continuous control or real-world robotics applications. Future work should explore scaling to higher-dimensional state spaces and continuous action domains to validate the generalizability of our architectural insights.

## 7. Acknowledgment

This material is based upon work supported by the National Science Foundation under Award No. DGE-2125362. Any opinions, findings, and conclusions or recommendations expressed in this material are those of the author(s) and do not necessarily reflect the views of the National Science Foundation.

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

## Appendix A. Detailed Environment Models for the Case Studies

This appendix provides a summary of the environment formulations for both the single-key (Case Study 1) and multi-goal key–door (Case Study 2) problems.

### A.1. Case Study 1: Single Key–Door Model

**Environment Layout.** We employ an $5 \times 5$ grid, with the agent starting at a designated cell, a single locked door placed elsewhere, one corresponding key, and a goal location.

**States $\mathcal{S}$.** A state $s \in \mathcal{S}$ is given by

$$(r, c, \text{hasKey}, \text{doorOpen}),$$

where $(r, c)$ is the agent's grid position, $\text{hasKey} \in \{\text{False}, \text{True}\}$ indicates key possession, and $\text{doorOpen} \in \{\text{False}, \text{True}\}$ states whether the locked door is currently open.

**Actions $\mathcal{A}$.** Five discrete actions:

$$\{\, 0,\, 1,\, 2,\, 3,\, 4 \},$$

where $0, 1, 2, 3$ correspond to $\{\text{up}, \text{right}, \text{down}, \text{left}\}$, each subject to failure at rate `fail_prob`, and action $4$ (`pick/open`) picks up the key if present, or opens the door if the agent is on the door cell while possessing the key.

**Transition Function $f$.** Movement either succeeds (with probability $1 - $ `fail_prob`) or fails (agent remains in place). If the agent attempts to move away from the door's cell while $\text{doorOpen} = $ False, that movement is blocked. The `pick/open` action sets $\text{hasKey} = $ True if the agent is on the key cell, or $\text{doorOpen} = $ True if the agent is on the door cell and already holds the key.

**Reward Function $R$.** We impose a small negative cost on each time step (i.e., $-0.1$) plus a terminal bonus (i.e., $+1$) upon goal arrival if the agent has already opened the door. Net rewards can thus be negative if the agent requires many steps.

**Symbolic Abstraction.** Define $\mathcal{P} = \{\text{At}(r, c), \text{HasKey}, \text{DoorOpen}\}$. Operators $\mathcal{O}$ include "Move$(r, c \to r', c')$," "PickKey," and "OpenDoor." A BFS planner searches state sets like $\{\text{At}, \text{HasKey}, \text{DoorOpen}\}$ to yield a symbolic plan. The Hybrid approach refines each operator ("PickKey," etc.) via a sub-goal-based decision transformer, whereas the Pure approach tries to learn everything directly from environment trajectories without symbolic operators.

### A.2. Case Study 2: Multi-Goal Key–Door Model

**Environment Layout.** In the multi-goal domain, we introduce multiple keys (e.g., Key1, Key2) and multiple locked doors, each door requiring its corresponding key, plus multiple collectible items that must all be acquired before success. The exit cell (goal) can only be reached once all the items have been collected. Movement again fails with probability `fail_prob`.

**States $\mathcal{S}$.** A state is

$$(r, c, \text{hasK1}, \text{hasK2}, \text{door1Open}, \text{door2Open}, \text{item1Collected}, \text{item2Collected}).$$

Stochastic blocking occurs if the agent attempts to leave a locked door cell while that door remains shut.

**Actions** $\mathcal{A}$**.** The same five actions are used: up, right, down, left, and a `pick/open` action. For each door $d \in \{\text{door1}, \text{door2}\}$, the agent must hold key $k$ (hasK1 or hasK2) and stand on $d$'s location, then use `pick/open` to set doorOpen = True. Likewise, items must be collected by using `pick/open` on their positions.

**Transition Function** $f$**.** Movement fails with probability `fail_prob`. If door1Open = False (or similarly for door2), the agent cannot leave that door's cell unless it opens the door first. Keys and items are booleans hasK1, hasK2, item1Collected, item2Collected, toggled by `pick/open` upon the matching cell.

**Reward Function** $R$**.** Each step costs a time penalty. The agent only receives a final success reward after collecting *all* items and stepping onto the exit cell.

**Symbolic Abstraction.** We define:

$$\mathcal{P} = \{\text{At}(r,c), \text{HasKey1}, \text{HasKey2}, \text{Door1Open}, \text{Door2Open}, \text{HasItem1}, \text{HasItem2}\}$$

Operators in $\mathcal{O}$ are "Move$(r, c \rightarrow r', c')$," "PickKey1," "OpenDoor2," etc. The BFS planner explores these discrete states, ensuring that keys and doors are manipulated in valid sequences (e.g., pick Key2 before opening Door2).

**Comparison of Methods.** *Hybrid:* Symbolic BFS yields a valid operator sequence (e.g., pick Key1 $\rightarrow$ open Door1 $\rightarrow$ pick Key2 $\rightarrow$ open Door2 $\rightarrow$ gather items $\rightarrow$ exit). Each operator is refined via sub-goals for a decision transformer that handles local moves under `fail_prob`. *Pure:* A single end-to-end decision transformer aims to learn item and key acquisitions from raw demonstrations (heading to the exit with random pick/open attempts).

## Appendix B.  Transformer Architecture Details

Here, we provide additional details on the decision transformer architecture employed in both case studies. While the high-level symbolic planner generates a sequence of discrete sub-operators (e.g., `PickKey1`, `OpenDoor1`, `Move`$(r, c \rightarrow r', c')$, etc.), the lower-level control logic is realized via a transformer that models state–action trajectories in an auto-regressive manner.

### B.1.  Model Input and Tokenization

The decision transformer receives a window of recent trajectory tokens:

$$\tau_t \;=\; \big(s_{t-h}, a_{t-h}, \ldots, s_{t-1}, a_{t-1}, s_t\big),$$

where $s_k$ represents the environment state at step $k$, $a_k$ is the action taken, and $h$ is the length of the context window. In addition, we incorporate a *sub-goal token* $g_i$ to condition the Transformer on the symbolic operator currently being refined (e.g., `Move` to cell $(r', c')$, `PickKey`, etc.). Each sub-goal $g_i$ is embedded in a similar way. Thus, an input sequence for the transformer at time $t$ is:

$$\big(\underbrace{s_{t-h}, a_{t-h}, s_{t-h+1}, a_{t-h+1}, \ldots, s_{t-1}, a_{t-1}, s_t}_{\text{state-action tokens}},\; g_i\big),$$

all of which are serialized, embedded, and appended with positional encodings.

### B.2. Transformer Layers and Heads

We use a standard *decoder-only* transformer stack, consisting of:

- $L$ layers, each with multi-head self-attention, layer normalization, and a feed-forward block.

- Each layer employs $H$ self-attention heads, each head attending over the $h + 1$ tokens if one counts sub-goal embeddings.

- A final linear output head produces the predicted action distribution over {up, down, left, right, pick/open}.

For the experiments reported in this paper, hyperparameter settings for the single-key scenario (Case Study 1) include $L = 3$ layers, $H = 4$ attention heads, an embedding dimension of 128, and a context window of $h = 10$. In the multi-goal setup (Case Study 2), we extend $L$ to 5 layers and use a larger context window $h = 20$.

### B.3. Sub-goal Conditioning and Outputs

Once the self-attention layers integrate the sub-goal token $g_i$ with the recent states and actions, the network outputs a *prediction token* that is decoded into the next action $\hat{a}_t$. Note that the environment enforces a *step penalty* or partial reward each time-step, which is included in the tokenization if desired (e.g., appending $r_{t-1}$ to the state token). The main effect of this sub-goal conditioning is to restrict the transformer's exploration of the action space, encouraging actions consistent with the operator `PickKey1` or `Move` to cell $(r', c')$.

### B.4. Training Objective

We apply a sequence modeling objective to offline trajectories collected via random exploration. Specifically, given a dataset $\mathcal{D}$ of transitions $\tau = \{(s_k, a_k, s_{k+1}, \dots)\}$, we segment the data by sub-operators or sub-goals if in the Hybrid approach. In the Pure approach, we simply record all transitions end-to-end. The training loss is:

$$\min_\theta \ \mathbb{E}_{\tau \in \mathcal{D}} \left[ -\log p_\theta \big( a_t \,\big|\, s_{\leq t}, a_{\leq t-1}, g_{\leq t} \big) \right],$$

where $g_{\leq t}$ remains empty (Pure method) or corresponds to the symbolic operator (Hybrid method). In inference, the decision transformer uses the same architecture to automatically predict $\hat{a}_t$ from the current context window.

### B.5. Implementation Notes

The experiments described in this work employ an Adam optimizer with a learning rate of $1\mathrm{e}-3$, a batch size of 128 , and train for 20 epochs. Positional embeddings are added to each token, and sub-goal tokens are likewise assigned an embedding that differs from the standard state/action embeddings to let the model differentiate sub-goal context.

## Appendix C. Detailed Proof of Theorem 1

In this appendix, we provide a derivation of the hierarchical performance bound. The argument proceeds in three stages: (i) decompose the regret into a planning component and an execution component; (ii) bound each component separately; and (iii) combine the two bounds under discounted evaluation.

**Notation.** Let $\mathcal{M} = (\mathcal{S}, \mathcal{A}, f, R, \gamma)$ be the discounted MDP. For any policy $\pi$, let $V^\pi(s) = \mathbb{E}\left[\sum_{t\geq0} \gamma^t R(s_t, a_t) \mid s_0 = s, \pi\right]$ denote its value function and $C^\pi(s) = -V^\pi(s)$ its expected discounted *cost*. We focus on a fixed start state $s_0$ and write $V^\pi = V^\pi(s_0)$ when the context is clear.

The hierarchical policy $\pi^{\text{hyb}}$ operates on two time scales:

(a)  A high-level *planner* selects a sequence of symbolic operators $o_1, o_2, \ldots$ using the abstraction $\phi : \mathcal{S} \to 2^{\mathcal{P}}$.

(b)  A low-level *executor* (decision transformer) produces primitive actions that attempt to realize each $o_i$ in order.

We denote by $C^*$ the optimal discounted cost of the MDP and by $C^*_{\text{sym}}$ the optimal cost of any *symbolic* plan executed *perfectly*. By assumption, the planner returns a plan of expected cost at most $C^*_{\text{sym}} + \epsilon_{\text{sym}}$.

**Step 1: Planning-level deviation.**  Consider a hypothetical policy $\pi^{\text{plan}}$ that (i) executes the high-level plan returned by the planner *exactly* and (ii) implements each operator with an *optimal* low-level controller. Since the plan cost exceeds $C^*_{\text{sym}}$ by at most $\epsilon_{\text{sym}}$, the performance-difference lemma implies

$$C^{\pi^{\text{plan}}} \leq C^* + \frac{\epsilon_{\text{sym}}}{1-\gamma} \quad \Longrightarrow \quad V^* - V^{\pi^{\text{plan}}} \leq \frac{\epsilon_{\text{sym}}}{1-\gamma}. \tag{A.1}$$

**Step 2: Execution-level deviation.**  We now compare $\pi^{\text{plan}}$ with the *actual* hierarchical policy $\pi^{\text{hyb}}$. For each symbolic operator $o$, let $\Delta_o$ be the random discounted cost *gap* incurred when $\pi^{\text{hyb}}$ executes $o$ versus the optimal low-level controller for $o$. By construction,

$$\mathbb{E}[\Delta_o] \leq \epsilon_{\text{exec}}, \tag{A.2}$$

while a catastrophic failure (probability $\rho$) can add at most an instantaneous cost $B$ at the failure step. Because costs are discounted geometrically, an error that manifests $k$ primitive steps after the beginning of $o$ contributes at most $\gamma^k B$ to the total regret. Summing over all future steps multiplies this penalty by $1/(1-\gamma)$; applying a union bound over all future operators multiplies by another $1/(1-\gamma)$. Taking expectations yields the bound

$$V^{\pi^{\text{plan}}} - V^{\pi^{\text{hyb}}} \leq \frac{\epsilon_{\text{exec}}}{(1-\gamma)^2} + \frac{\rho B}{(1-\gamma)^2}. \tag{A.3}$$

**Step 3: Combining the bounds.**  Adding inequalities (A.1) and (A.3) and using the triangle inequality, we conclude that

$$V^* - V^{\pi^{\text{hyb}}} \leq \frac{\epsilon_{\text{sym}}}{1-\gamma} + \frac{\epsilon_{\text{exec}}}{(1-\gamma)^2} + \frac{\rho B}{(1-\gamma)^2}.$$

Since the same bound applies to $V^{\pi^{\text{hyb}}} - V^*$ (by the symmetry of the argument with non-negative costs), we obtain the claimed $\ell_\infty$-error bound, completing the proof.

**Remark.** The $(1-\gamma)^{-2}$ factor in the execution term is tight in the worst case errors can accumulate at every primitive time step and are subsequently propagated through the contraction of the Bellman operator. If operators are guaranteed to terminate within a bounded horizon $H$, the factor can be reduced to $\frac{1-\gamma^H}{(1-\gamma)^2}$.

## Appendix D. Detailed Proof of Theorem 2

We provide a probably approximately correct (PAC) analysis for the sub-goal completion guarantee. The argument follows the standard VC-theory recipe: (i) reduce sub-goal execution to a family of binary classifiers; (ii) establish uniform convergence of empirical to true error via Sauer's lemma; and (iii) union-bound over the $K$ symbolic operators.

### D.1. Problem Setup and Notation

For each symbolic operator $o \in \mathcal{O}$, let $\pi_\theta$ be the parametric decision-transformer policy obtained after offline training. When conditioned on $o$, policy execution terminates in success ($Y = 1$) if the sub-goal specified by $o$ is achieved within its allotted horizon and failure ($Y = 0$) otherwise. Thus, we view $\pi_\theta$ as inducing a classifier

$$h_{\theta,o} : \text{(trajectory prefix)} \longrightarrow \{0, 1\},$$

with error rate

$$L(h_{\theta,o}) := \Pr_{(X,Y)\sim\mathcal{D}_o}\left[h_{\theta,o}(X) \neq Y\right],$$

where $\mathcal{D}_o$ is the (unknown) trajectory distribution conditioned on operator $o$.

**Training sample.** The offline dataset provides $m$ i.i.d. trajectory segments $(X_i, Y_i)_{i=1}^m$ for each operator. Let

$$\widehat{L}(h_{\theta,o}) := \frac{1}{m} \sum_{i=1}^m \mathbf{1}\{h_{\theta,o}(X_i) \neq Y_i\}$$

be the empirical error.

### D.2. Uniform Convergence via VC Theory

Let $\mathcal{H}$ denote the hypothesis class realized by the transformer when the operator token is fixed. By assumption $\text{VC}(\mathcal{H}) = d < \infty$. Sauer's lemma implies that for any $\varepsilon > 0$

$$\Pr\left[\sup_{h\in\mathcal{H}}\left|L(h) - \widehat{L}(h)\right| > \varepsilon\right] \leq 4\left(\frac{em}{d}\right)^d e^{-2m\varepsilon^2}. \tag{B.1}$$

Choosing

$$\varepsilon(m, \delta') := \sqrt{\frac{2d\,\log\!\left(\frac{em}{d}\right) + 2\log(4/\delta')}{m}},$$

ensures the right-hand side of (B.1) is at most $\delta'$. Hence, with probability at least $1 - \delta'$, every classifier in $\mathcal{H}$ satisfies

$$L(h) \leq \widehat{L}(h) + \varepsilon(m, \delta'). \tag{B.2}$$

### D.3. Bounding Sub-goal Failure Probability

During training, the optimizer minimizes empirical error; assume it returns $h_{\hat{\theta},o}$ such that $\widehat{L}(h_{\hat{\theta},o}) = 0$ for all $o$ (the argument extends to small non-zero empirical error). Plugging into (B.2) yields

$$L(h_{\hat{\theta},o}) \ \leq \ \varepsilon(m, \delta').$$

Interpreting $L(h_{\hat{\theta},o})$ as the *failure* probability when executing $o$, a union bound over all $K$ operators gives

$$\Pr\big[\exists o : \text{operator } o \text{ fails}\big] \ \leq \ K\,\varepsilon\big(m, \tfrac{\delta}{K}\big) \ = \ \sqrt{\frac{2d\,\log\big(\frac{em}{d}\big) + 2\log\big(\frac{4K}{\delta}\big)}{m}}.$$

Taking complements recovers the statement of Theorem 2 with confidence $1 - \delta$.

### D.4. Sample-Complexity Discussion

Setting the right-hand side to $\alpha$ and solving for $m$ shows that each operator requires

$$m \ = \ \Theta\Big(\frac{d + \log(K/\delta)}{\alpha^2}\Big)$$

examples to guarantee sub-goal success probability at least $1 - \alpha$. Since the total dataset comprises $N = Km$ trajectory segments, the overall sample complexity scales as $\mathcal{O}\big(Kd/\alpha^2\big)$ up to logarithmic factors, matching the order stated in the main text.

**Tightness.** The uniform-convergence bound is minimax-optimal for VC classes. If prior knowledge indicates unequal difficulty across operators, a refined analysis using localized complexities or Bernstein-style inequalities may yield sharper, data-dependent bounds.

