# OpenReview forum: "Hierarchical Neuro-Symbolic Decision Transformer"
_nesyconf.org/NeSy/2025/Conference_Phase_2 — NeSy 2025 - Phase 2 Poster_

### Official Review · Reviewer_wbAG · 2025-07-06
**interesting paper on hierarchical hybrid reinforcement learning**

**Rating:** 7
**Confidence:** 2

**Review:**

Paper summary:
The manuscript proposes a novel architecture that unifies symbolic planning with a transformer-based policy, thereby enabling high-level logical task decomposition alongside low-level control. Through numerical evaluations, the paper shows that the proposed approach outperforms purely end-to-end neural baselines, achieving higher success rates and improved sample efficiency in tasks with long-horizon dependencies. Evaluation is performed on two toy problems of different complexities and compared against purely neural and purely symbolic methods, outperforming both.
Proposed method:
The proposed method is composed of a decision transformer, which reframes policy learning as a sequence modeling problem and imitates offline trajectories that demonstrate desired behaviors, and a symbolic planner. The overarching goal is to use the global logical consistency afforded by symbolic planners while exploiting the representational and sequence-modeling capabilities of a large-scale neural network for local control and refinement. The planner solves a combinatorial optimization problem yielding a sequence of high-level actions, which is then translated into a low-level action plan by a decision transformer conditioned on each sub-goal in the sequence, producing a policy for the next low-level action until the sub-goal is declared achieved.

Strenghts:
S1. Interesting paper providing a viable way of mixing a high-level symbolic planner with low-level localized neural action prediction for long-term planning.
S2. Provides both theoretical bounds and empirical performance estimates against several baselines, showing performance benefits in terms of Success rate and Sample Efficiency.
S3. Well written and organized paper, appendix provides full experimental set up.
Weaknesses:
W1. Experiments performed on ad-hoc relatively simple benchmarks.
W2. Not clear how the symbolic abstraction step, taking as input the environment s_t, (light green box in Figure 1) is handled in the selected problems, or could be handled in more complex ones. In the current experiments, it appears that the solution relies on a predefined set of symbol states, that may limit the applicability of the proposed techniques in more complex settings.
Additional remarks
-	I found it interesting that pure decision transformer, that should be SOTA, while very efficient, have a much lower success rate than other neural architectures, such as GRU or LSTM
-	A more in-depth discussion of the failure cases could be of interest

**Anonymity:**

Remain anonymous

---

### Official Review · Reviewer_DhDp · 2025-07-08

**Rating:** 7
**Confidence:** 2

**Review:**

The paper introduces a hierarchical neuro-symbolic decision transformer.
This method combines high-level planning with a low-level transformer-based policy.
The method is validated in a grid-world environments.
The authors propose a bidirectional mapping between discrete symbols and continuous sub-goals.
The authors perform an extensive evaluation of the method using multiple methods including multiple recurrent neural networks, pure decision transformers, and traditional reinforcement learning methods.

* The paper is well-written and the results are promising.
* The combination of high-level planning and low-level policy learning is a very interesting open problem.
* One open question I have is how does the selection of the high-level and low-level actions be automatically defined in more complex environments?
* As mentioned by the authors, extending this method to more complex environments would be interesting future work.
Minor details:
* Abstract has a typo. It has a double comma after "purely neural".
* A more detailed description of the caption in Figure 1 would be beneficial.

**Anonymity:**

Remain anonymous

---

### Official Review · Reviewer_z8Zf · 2025-07-09

**Rating:** 4
**Confidence:** 3

**Review:**

## Strengths

**Intuitive approach.** The proposed method is an intuitive combination of symbolic planning and transformers.

**Relevance**. This work is certainly relevant to the venue.

**Sentence-level writing.** While I have some concerns about clarity, the writing at the sentence-level is good. I only noticed a few small typos.

**Theoretical and empirical results.** While I have some concerns about the results themselves, it is notable that the paper includes both theoretical and empirical results.

## Weaknesses

**Novelty.** The related work section is very broad, which is good, but it is lacking depth and missing relevant references, e.g., \[1-5\]. Moreover, the proposed combination of symbolic planning and neural execution is something that has been considered quite a bit, and it’s not completely clear why the proposed approach is better than or different from existing ones. The proposed approach also has shortcomings that related works do not have, which I will note below.

\[1\] Learning neuro-symbolic skills for bilevel planning. Silver et al., CoRL 2022\.
\[2\] Vq-cnmp: Neuro-symbolic skill learning for bi-level planning. Atkas et al., 2024\.
\[3\] Symbolic plans as high-level instructions for reinforcement learning. Illanes, et al. AAAI 2020\.
\[4\] The Logical Options Framework. Araki et al, ICML 2021\.
\[5\] Learning Temporally Extended Skills in Continuous Domains as Symbolic Actions for Planning. Achterhold et al., CoRL 2022\.

**Clarity & missing problem formulation.** The paper is missing a Problem Formulation section. The formulation can be partially inferred, but there is still some ambiguity about what needs to be given in order to apply the approach. The details about the approach are also unclear. For example:

* How and when exactly is re-planning performed? What measures are taken (if any) to avoid possible infinite loops, where the same plan is returned and then immediately fails?
* How exactly are trajectories segmented?
* How exactly is tokenization performed? What must be assumed about the observations, actions, and subgoals?

**Technical concerns about the approach.** I have the following concerns about the approach itself:

1. The approach requires a mapping from operators to subgoals. First, it seems limiting that this mapping assumes a unique subgoal for each operator. The intended effect of a general skill like “pick” should have different subgoals depending on the context---for example, different grasps of an object. Second, the relationship between subgoals and abstract states is not clear. For example, why do we need to introduce new subgoals at all, rather than using the effects of the operators directly?
2. It’s not clear to me what level of generalization, if any, can be expected. In principle, it seems like the approach should be able to generalize to different goals and low-level initial states. But I don’t know if the approach can generalize over objects, which is something that other neuro-symbolic approaches are able to do.
3. The approach requires quite a lot of prior knowledge in different forms, including a potentially large dataset. At this point, it might be easier to just hand-code solutions.

**Technical concerns about the results.** I have the following concerns about the results:

1. Theorem 1 makes strong assumptions and has fairly obvious conclusions.
2. Both theorems are pretty far removed from the particular approach, e.g., they have nothing to do with transformers or symbolic operators.
3. Some terms in the theorems are not defined, e.g., $C\_i$ in Theorem 1 and $e$ and $m$ in Theorem 2\.
4. I am not confident about this, but I have a detailed concern about Theorem 2\. There seems to be an implicit assumption that the subproblems are independent. But that is certainly not the case. For example, if there are multiple low-level states that achieve a subgoal, then the distribution of low-level states reached by an “upstream” skill influences the distribution of the “downstream” skill.
5. The empirical results are quite limited in terms of the complexity and breadth of the environments considered.
6. I don’t understand why the symbolic baseline is ever “beat” by the main approach. These environments seem very simple and I would imagine it is straightforward to code a perfect symbolic baseline.
7. It’s not clear whether there is variation of any kind in the environments, or any attempt to test generalization between training and test.
8. I don’t think that Figure 5 adds much beyond Figure 4\.

**Anonymity:**

Remain anonymous